# Effect of Phase Transformations of a Metal Component on the Magneto-Optical Properties of Thin-Films Nanocomposites (CoFeZr)_x_ (MgF_2_)_100−x_

**DOI:** 10.3390/nano11071666

**Published:** 2021-06-24

**Authors:** Elena Alexandrovna Ganshina, Vladimir Valentinovich Garshin, Ilya Mikhailovich Pripechenkov, Sergey Alexandrovich Ivkov, Alexander Victorovich Sitnikov, Evelina Pavlovna Domashevskaya

**Affiliations:** 1Department of Magnetism, Faculty of Physics, Lomonosov Moscow State University, Leninskiye Gory B 1-2, GSP-1, 119991 Moscow, Russia; eagan@mail.ru (E.A.G.); irving.lambert@mail.ru (V.V.G.); priil@yandex.ru (I.M.P.); 2Department of Solid State Physics and Nanostructures, Faculty of Physics, Voronezh State University, Universitetskaya pl. 1, 394018 Voronezh, Russia; ivkov@phys.vsu.ru; 3Department of Solid State Physics, Faculty of Radio Engineering and Electronics, Voronezh State Technical University, 14 Moskovsky Ave., 394026 Voronezh, Russia; myftt@yandex.ru

**Keywords:** nanocomposites, magneto-optical properties, transversal Kerr effect, structural phase transitions, magnetic percolation threshold, self-organization processes

## Abstract

The results of complex studies of structural-phase transformations and magneto-optical properties of nanocomposites (CoFeZr)_x_ (MgF_2_)_100−x_ depending on the metal alloy content in the dielectric matrix are presented. Nanocomposites were deposited by ion-beam sputtering onto glass and glass-ceramic substrate. By studying the spectral and field dependences of the transversal Kerr effect (TKE), it was found that the transition of nanocomposites from superparamagnetic to the ferromagnetic state occurs in the region of *x**_fm_***~30 at%, that corresponds to the onset the formation of ferromagnetic nanocrystals CoFeZr with hexagonal syngony in amorphous dielectric matrix of MgF_2_. With an increase of concentrations of the metal alloy for *x > x**_fm_***, the features associated with structural transitions in magnetic granules are revealed in the TKE spectra. Comparison of the spectral and concentration dependences of TKE for nanocomposites on the glass and glass-ceramics substrates showed that the strongest differences occur in the region of the phase structural transition of CoFeZr nanocrystals from a hexagonal to a body-centered cubic structure at *x* = 38 at.% on the glass substrates and at *x* = 46 at.% on glass-ceramics substrates, due to different diffusion rates and different size of metal nanocrystals on amorphous glass substrates and more rough polycrystalline glass-ceramics substrates.

## 1. Introduction

The continuing interest in nanocomposite (NC) materials is due to the modification of the known metals and alloys properties during their transition to the nanocrystalline state. Magnetic nanocomposites of metal-dielectric type have a number of unique physical properties that are promising for their use in spintronics, in the technologies for recording and storing information, in the creation of shielding coatings, sensitive magnetic sensors and other devices [1,2,3]. The object of intensive experimental and theoretical research is the influence of the phase composition and microstructure of nanocomposites, which are realized in the process of self-organization of atoms during their fabrication, on the magnetic, magneto-transport and magneto-optical properties of nanomaterials [4].

All these properties of nanocomposites appreciably depend on their composition and microstructure, especially on the shape and size of metal granules, on their distribution over the sample volume, and primarily on the concentration of the magnetic metal phase. In this regard, magneto-optical (MO) studies are of a considerable interest, since they are sensitive to the changes in the magnetic and electronic structures caused by the characteristic sizes, shapes, and topology of metal particles [5,6,7,8,9,10].

Concentration dependences of the electromagnetic properties of nanocomposites show dramatic nonlinear changes at the threshold of percolations *x***_per_**, i.e., at such value of the concentration of metal component at which a finite conducting network of contacting metal particles is formed in the entire volume of the sample. It can be stated that the threshold of percolation is an intermediate state during the transition of the nanocomposite from an electrically nonconducting to an electrically conductive state, where the metal granules begin to come into contact with each other. In this area, all the unique physical properties that are inherent to granular composites are more pronounced [11].

With decreasing metal concentration *x*, from *x* = 100 at.% down to the percolation threshold *x*_per_, the resistivity of nanocomposites increases, as physical contact between most of the granules continuously disappears. There is still the tunneling conductivity for the concentrations down to *xc* (*x_per_ > x > x_c_*), where metal–insulator transition occurs. In this concentration range, novel transport properties, such as tunnel anomalous Hall effect and logarithmic temperature dependence of conductivity, have been found [12]. Hopping conductivity of different types as well as co-tunneling was observed in dielectric regimes below *xc* [11,13,14].

The character of magnetic properties in NC is even more complex. In the region of very low concentrations of the metal component (*x* << *x***_per_**) or large (*x* > *x***_per_**) NC exhibits superparamagnetic or ferromagnetic (FM) properties, respectively. The magnetic percolation threshold *x_fm_* does not coincide with the transport threshold [15], it is always less than *x***_per_**, since the ferromagnetic order (FM) can occur without physical contact between the granules. Therefore, it is possible to define *x_fm_* as the concentration at which the magnetization and the coercive force dramatically increase. Depending on the relative intensity of the dipole-dipole and exchange interactions, complex magnetic states can occur in the vicinity *x_fm_* of superferromagnetic (SFM) state, superspin glass state, superparamagnetism (SPM), and mixed SFM–FM–SPM state [13,15]. Moreover, the presence of magnetic ions between the granules can enhance the FM exchange between them, or vice versa, promote antiferromagnetic exchange [14,16], which makes various scenarios possible. It was shown in [17] that magneto-optical methods allow detecting the presence of SFM states in nanocomposites.

As a magnetic component of film composites, ferromagnetic metals Fe, Co, Ni and alloys based on them, which are amorphized by boron or zirconium additives, are rather suitable.

Most of the known metal-insulator composite systems are obtained using SiO_2_, Al_2_O_3_, MgO, and ZrO_2_ oxide dielectrics. At the same time, due to the presence of oxygen in the condensed medium, a significant oxidation of the surface-metal granules occurs, which affects the magnetic and MO characteristics of the composites.

Recently, the studies of the composites with oxygen-free matrices have been reported. For example, in [18], while studying the magnetotransport properties of nanocomposites with Fe or Fe_51_Co_49_ alloy distributed in the MgF_2_ dielectric matrix, a giant magnetoresistance was found at room temperature.

Nanocomposites with a carbon matrix were also studied. In [19], the electrical properties of metal-carbon composites and magneto-optical properties [20] were investigated, showing that self-organization processes in the preparation of nanocomposites with a carbon matrix differ from the self-organization processes in nanocomposites with oxide matrices. A detailed study of the atomic structure, phase formation, and substructure of Co nanocrystals and Co_45_Fe_45_Zr_10_ alloy in granular nanocomposites with MgF_2_ dielectric matrix, depending on the ratio of the metallic and dielectric components, was carried out in [21,22] using X-ray diffraction (XRD), X-ray electron spectroscopy (XPS), and infrared spectroscopy (IR). 

It is well known that magneto-optical spectroscopy can give unique information about magnetic microstructure of the both crystalline and amorphous ferromagnetic metals and for magnetic composites [5,7,9] However, to the best of our knowledge, there are no joint studies of the structural and magneto-optical properties of granular systems in the oxygen-free matrix.

The aim of this work is to study magneto-optical properties of nanocomposites (CoFeZr)_x_ (MgF_2_)_100−x_ with a multi-element ferromagnetic alloy in oxygen-free dielectric matrix of MgF_2_ on the glass and glass-ceramics substrates and to identify the influence of the composition and phase transformations on their magneto-optical properties and self-organization processes.

Thus, for the first time, new data will be presented on the magneto-optical properties and the effect of phase transitions on them when changing the composition of a composite obtained of a three-element alloy CoFeZr and an oxygen-free matrix MgF_2_ and the MO percolation threshold will be determined. 

## 2. Objects and Methods of the Study

The main known methods for producing thin composite films are thermal, cathode, and ion-plasma sputtering. Each of these methods has its own advantages and disadvantages, depending on the sputtered material and the goal of the work. In [4], the analysis of existing methods for producing nanocomposites was carried out and it was concluded that the most universal method is ion-plasma magnetron sputtering, when the separation of the condensed medium into two components (dielectric and metal) is realized as a result of their self-organization in a single deposition process.

This paper presents the results of the study for two series of nanocomposite films (CoFeZr)_x_ (MgF_2_)_100−x_ with metal granules of the CoFeZr alloy in the MgF_2_ dielectric matrix on glass and glass-ceramics substrates within the range of changes in the nominal values of *x* calculated from the geometry of the composites deposition, *x* = 9–51 at.%, including the percolation threshold [4].

Nanogranulated composite films with different content of the initial metal alloy Co_45_Fe_45_Zr_10_ and the dielectric component MgF_2_ were obtained at the original ion-beam sputtering facility [4,23]. The optimal option of providing the identical conditions for obtaining film samples is the formation of composites with different ratios of the dielectric and metal phases simultaneously in one deposition process. This option is implemented when sputtering a composite target with an uneven and asymmetric placement of the metal and dielectric parts, and as a result a given concentration gradient is formed in the sputtered material in a single cycle [23].

Sputtering of a composite target made from a plate of an amorphous metal alloy Co_45_Fe_45_Zr_10_ with inserts made of dielectric MgF_2_ material was carried out in an argon atmosphere at the operating pressure of ~5 × 10^−4^ Torr. 

An alloy target of the composition Co_45_Fe_45_Zr_10_ was used to precipitate the amorphous ferromagnetic metal component of the composites. It was made by vacuum melting from metals of the appropriate composition using an induction furnace. Preparation of attachments for the alloy was carried out from carbonyl high-purity iron (99.9%), high-purity cobalt (99.98%) and zirconium (99.8%) with a weight content of components in accordance with the composition of the alloy Co_45_Fe_45_Zr_10_. The melt of the corresponding composition was poured into a specially prepared ceramic mold in a vacuum. The target had sizes of 270 mm × 70 mm × 14 mm. The target were ground on both sides, soldered to a water-cooled base, and placed in the spray position in the ion-beam sputtering facility. MgF_2_ insets with a thickness of ~2 mm and a width of ~9 mm were fixed on the surface of the alloy target perpendicular to its longitudinal axis, not equidistant, in order to obtain a continuous set of concentrations of the components of the composite target along its length in one technological deposition cycle (Figure 1).

For the production of dielectric inserts, magnesium fluoride MgF_2_ for optical ceramics, doped with calcium, pure (GOST 6-09-01-731-91) with a molecular weight of 62.30 atomic mass units was used. The mass fraction of calcium (Ca) was about 0.1–0.3%.

Deposition was performed on glass-ceramics and glass substrates with a size of 60 mm × 48 mm. The use of non-uniform arrangement of dielectric inserts allowed us to obtain a continuous spectrum of changes in the compositions of alloy and dielectric components of the composite from 20 to 70 at.% of the metal phase in one technological cycle of deposition. The approximate ratio of the volume fraction of an alloy to the volume fraction of a dielectric with a uniform distribution of attachments can be estimated by the formula:$$\frac{V_{F}}{V_{D}}=\frac{R_{F}}{R_{D}}\cdot \frac{S_{F}}{S_{D}}=\frac{R_{F}}{R_{D}}\cdot \frac{S-ns}{ns}=\frac{R_{F}}{R_{D}}\cdot \frac{L-nb}{nb}$$
where: *R_F_* and *R_D_*-coefficients spraying metal alloy and dielectric, respectively; *S_F_* and *S_D_* sputtering target area occupied by the alloy and dielectric, respectively; *S* is the area of the entire target (270 mm × 70 mm); *n*-the number of dielectric inserts in composite target; *s*-area of a single insert (9 mm × 70 mm); *L*-length of the target compound (270 mm); *b*-width of the dielectric insert. 

The layer thicknesses of the film samples of granular composites with different compositions varied smoothly in the range of 2–4 μm while increasing *x* in the alloy composition [22].

The concentrations of the chemical elements that make up the composites were measured by electron-probe X-ray spectral microanalysis with JEOL JSM-6380LV scanning microscope (Japan) attachment, equipped with three crystal diffraction spectrometers and an energy-dispersion analysis system, with an error less than 1.0 at.% of the content of the measured element. The determined composition of the samples represented the content of each chemical element in the composite layer (CoFeZr)_x_ (MgF2)_100−x_, expressed in atomic percentages. In this case, the variable x of the value of metal component in the composite is the sum of the atomic concentrations of the three metals of the alloy Co, Fe, and Zr.

The phase composition of nanocomposites and nanocrystals formed as a result of self-organization of two components, metallic and dielectric in different ratios, were studied by X-ray diffraction (XRD) with DRON-4 diffractometer (St. Petersburg city, Russia) using Co Kα radiation. X-ray diffraction patterns were obtained in the step-by-step scanning model [21,22].

The magneto-optical (MO) properties of nanocomposites were studied in the geometry of the transversal Kerr effect (TKE), which consists in changing the intensity of *p*-polarized light reflected by a sample when magnetized in a magnetic field parallel to the sample surface and perpendicular to the plane of light incidence. The TKE value,
*δ* = [*I*(*H*)–*I*(–*H*)]/2*I*(0),
where *I*(*H*) and *I*(0) are the intensities of reflected light in the presence and absence of a magnetic field, respectively, has been measured by the dynamic method at the magnetic field modulation. This method allows the use of a differential measurement scheme, providing the accuracy measuring the relative changes in the reflex light intensity of ~10^−5^ [20,24].

The TKE spectra were recorded in the energy range *E* = 0.5–4.0 eV at the applied magnetic field up to 3.0 kOe and room temperature. Magnetic field TKE(H) dependences were measured at some fixed energies to analyze the magnetic structure of the nanocomposites.

The magnetic properties of the samples were studied on a LakeShore Vibrating Sample Magnetometer (VSM) 7407 in fields up to 16 kOe.

## 3. Structural-Phase Transformations in the Composites Depending on the Metal Alloy Content in the Dielectric Matrix

Figure 2 shows diffraction patterns of nanocomposites with different compositions (CoFeZr)_x_ (MgF_2_)_100−x_ obtained on the glass substrates.

Nominal content of the metal component *x* in at.% for the samples is indicated to the right of each diffraction pattern. The lower diffraction patterns in Figure 2 of polycrystalline cobalt of the hexagonal modification ᾳ-Co and polycrystalline iron of the body-centered cubic (BCC) modification ᾳ-Fe, as well as the upper diffraction pattern of MgF_2_ powder, are given for comparison.

From the analysis of XRD data, it follows that in the range of low concentrations (*x* ≤ 25 at.%) the metal alloy CoFeZr is in the X-ray amorphous state in the form of metal clusters distributed in MgF_2_ nanocrystalline matrix, represented in diffraction patterns for the values *x* = 9–30 at.% by two wide reflections (110) and (220) at 2θ = 32° and 2θ = 67° from MgF_2_ nanocrystals.

But just at *x* = 30 at.% the halo appears in the area of angles 50° < 2θ < 60°, indicating at the beginning of formation of the CoFeZr nanocrystals, and at *x* = 34 at.%, the first three reflections appear, indicating at the formation of hexagonal nanocrystals based on the α-Co structure, mainly oriented in the plane of the base of hexagonal lattice (002), that is especially clearly visible at *x* = 38 at.%.

However, with a further increase in the metal alloy content, starting from *x* = 43 at.%, the crystal structure of CoFeZr nanocrystals is rearranged into a cubic volume-centered phase based on α-Fe with a predominant orientation of (110), like that of α-Fe. And already at *x* = 47–50 at.% in the amorphous MgF_2_ matrix (its reflections (110) and (220) disappeared at *x* = 38 at.%), only one nanocrystalline BCC phase of the CoFeZr alloy is present in the composites.

Figure 3 shows diffraction patterns of the samples with different compositions obtained on glass-ceramics substrates. The upper diffraction pattern were obtained from MgF2 powder. This figure also shows the initial alloy Co_45_Fe_45_Zr_10_ on the glass-ceramics substrate without MgF_2_, which provides an intense reflection in the form of halo in the range of angles 2θ = 45–60° with a small peak at the top position close to the position of the most intense lines (110) of α-Fe and the second by intensity of the line (002) α-Co, and with the most intense lines of nanocrystals CoFeZr in the composites with *x* = 41–51 at.%.

Hence it follows that the film of the pure alloy Co_45_Fe_45_Zr_10_ on the glass-ceramics, along with the amorphous phase of CoFeZr, can possibly contain nuclei of clusters of both cubic and hexagonal symmetry. The remaining weak lines on this diffraction pattern are attributed to glass-ceramics substrate.

When comparing the diffraction patterns of the dielectric component MgF_2_ and the glass-ceramics substrate, one can see that all of the main lines of MgF_2_ coincide with the lines of the glass-ceramics having a larger number of reflections. And since glass-ceramics, unlike glass, is not X-ray amorphous, it provides additional weak reflections in all of the diffraction patterns of nanocomposites with different composition.

However, from Figure 3 it follows that at low concentrations(*x* ≤ 28 at.%) the metal alloy CoFeZr is in the X-ray amorphous state in the form of metal clusters distributed in the MgF_2_ nanocrystalline matrix, represented in the diffraction patterns of Figure 3 by two main reflections (110) and (220) at the angles 2θ = 32° and 2θ = 67°. All other weak reflections are attributed to the glass-ceramics substrate.

For the composition of the NC with *x* = 35 at.%, the first reflection (100) from the group of three ones appears in the diffraction pattern in the halo region, corresponding to the beginning of the formation of hexagonal nanocrystals of the CoFeZr alloy, with the most intense line (100), in contrast to the metal cobalt with the most intense line (101).

But at *x* = 46 at.% the most intense lines from the group of three ones becomes the middle line, which at *x* = 51 at.% remains the dominant one in this range of Bragg angles, like α-Fe, as a result of the rearrangement of the hexagonal crystal structure of the alloy into a BCC one.

A careful comparison of the XRD results for the composites of two series (Figure 2 and Figure 3) obtained on amorphous glass substrates and on polycrystalline glass-ceramics substrates shows that the substrate material affects self-organization processes and structural-phase transitions caused by a change in the ratio of metal and dielectric components in nanocomposites. So, if the beginning of transition from the amorphous CoFeZr alloy to the nanocrystalline hexagonal phase on both types of the substrates occurs at very close compositions of about *x* ≈ 30 at.%, then the compositions for the beginning of the second phase transition of the CoFeZr alloy from the hexagonal to the BCC structure differ significantly: *x* = 38 at.% in nanocomposites on the glass substrates and *x* = 46 at.% in nanocomposites on glass-ceramics substrates. At the same time, in the latter, the phase transition process is not completed within the observed limits of concentrations, and at *x* = 51 at.%, the traces of two lines (100) and (101) of the hexagonal phase are still observed by both sides of the main line (110) of the BCC phase of CoFeZr nanocrystals. 

In addition, the sizes of the alloy nanocrystals (according to Scherrer) also differ, and they are about 15–20 nm on a glass substrate and about 25–30 nm on a glass-ceramics one, with their same orientation relative to the substrates [22,23].

## 4. Results of MO Investigations and Their Discussion

### 4.1. MO Properties of Nanocomposites (CoFeZr)_x_ (MgF_2_)_100−x_ on Glass Substrates

X-ray diffraction and magneto-optical investigations were performed with the same samples.

When studying magneto-optical properties, the spectral, field, and concentration dependences of the transverse Kerr effect were obtained.

Figure 4 shows the spectral and field dependences of the TKE for the samples on a glass substrates with metal phase concentrations *x ≤* 30 at.%.

In this concentration range, the behavior of the TKE spectra is similar to those ones of nanocomposites with metal granules CoFeZr in oxide matrices [9]. With an increase of the metal concentration, the TKE module value increases and attains the maximum value of the negative effect for the sample with *x_c_* = 30 at.% at the energy of approximately ~1.75 eV.

For the samples with *x* ≤ 20 at.% a linear dependence of the TKE on the magnetic field is observed. With an increase of the concentration up to *x* = 30 at.% there is a nonlinear dependence of TKE on the magnetic field, but the value of TKE does not reach saturation in the fields up to 3kOe.

For a more complete understanding of the magnetic structure of our films, we also carried out magnetostatic studies on a vibration magnetometer in fields up to 16 kOe. Figure 5 shows the hysteresis loops and the dependence of coercive force H_c_ on the concentration of the metal component for nanocomposites (CoFeZr)_x_ (MgF_2_)_100−x_ on the glass substrates.

Analysis of the hysteresis loops showed that the value of the coercive force H_c_ in all of the composites at *x* ≤ 30 at.% is equal zero. Therefore, the main contribution to the coercive force is made by single superparamagnetic particles, and after that, a nonlinear increase in the coercive force is observed, which is explained by the formation of FM clusters in the sample.

Thus, in the concentration range *x* ≤ 30 at.%, the samples are superparamagnets, and the critical magnetic percolation threshold for this system is *x_fm_* = 30 at.%.

When the concentration increases above magnetic percolation threshold *x_fm_* = 30 at.%, some features in the behavior of magneto-optical properties are observed that correlate with the features of the structural properties of nanocomposites, namely, with the beginning of the nucleation of CoFeZr nanocrystals of hexagonal syngony.

Figure 6 shows the spectral and field dependences of the TKE samples on the glass substrates in the region of metal concentrations *x* ≥ 30 at.%. According to the given curves it follows that at *x* = 34 at.%, the type of the TKE spectra changes. The effect begins to decrease in the energy range E > 1.57 eV, and the peak is shifted to E~1.65 eV. The shift of the TKE maximum coincides with the appearance of hexagonal CoFeZr nanocrystals in the amorphous dielectric matrix. TKE field dependencies become more magnetically hard.

Further increase of the metal concentration at *x* = 38 at.% leads to a sharp decrease in the magnitude of the MO effect and a further shift of TKE maximum towards lower energies. This behavior can be attributed to the nucleation of BCC nanocrystals based on α-Fe. In the same concentration region, a reflection corresponding to α-Fe (110) begins to appear in the diffraction pattern. In the sample with *x* = 43 at.%, the effect in TKE spectra begins to increase once again, and a maximum appears in the region of 1.25 eV, which is manifested in the diffraction pattern as the growth of intensity of one of the most intense lines (110), as in the alloy of CoFe and α-Fe.

With increasing x to 47–51 at.%, a further transformation of the shape of the MO spectra occurs: the modulus of the TKE value decreases, and the maximum of the spectrum shifts towards lower energies.

Figure 7 shows the concentration dependences of the TKE for nanocomposites (CoFeZr)_x_ (MgF_2_)_100−x_ at the energy values of 1.14 eV, 1.97 eV, and 3.17 eV. Unlike of NCs with amorphous metallic granules in oxide matrices, where the maximum on the concentration dependences of TKE corresponding to the threshold of magnetic percolation was observed in the near-IR region of the spectrum [9], in the studied nanocomposites with MgF_2_ matrix, this maximum at *x**_fm_*** = 30 at.% is most pronounced in the visible region of the spectrum at the energy values of 1.97 eV and 3.17 eV.

A more complex behavior of the concentration dependences of the TKE is observed in the near-IR region of the spectrum.

At the energy of 1.14 eV, we can distinguish an inflection in the *x**_fm_*** region and two maxima corresponding to the structural-phase transition from the amorphous phase to the nanocrystalline hexagonal phase (*x* = 34 at.%) and a feature in the concentration range of 38 ≤ *x* ≤ 47 at.%, at which the crystal structure of CoFeZr is rearranged from a hexagonal to a body-centered cubic lattice.

Thus, the concentration dependences of the TKE correlate well with the results of X-ray diffraction studies of structural-phase transitions in nanocomposites (CoFeZr)_x_ (MgF_2_)_100−x_, on the glass substrates.

### 4.2. MO Properties of Nanocomposites (CoFeZr)_x_ (MgF_2_)_100−x_ on the Glass-Ceramics Substrates

Figure 8 shows the spectral and field dependences of the TKE in the samples of (CoFeZr)_x_ (MgF_2_)_100−x_ on the glass-ceramics substrates at different concentrations of the metallic phase.

As well as for NC on the glass substrates, the modulus of the TKE value in the samples on glass-ceramics substrates increases with an increase of the concentration up to values of *x_fm_* = 31.4 at.%.

Magnetic measurements (Figure 9) also showed that at the concentrations of *x* = 31.4 at.% coercive force of H_c_ is zero, and non-zero values of H_c_ appear at *x* > 31at.%. Comparison of the dependences of coercive forces for the systems on glass and glass-ceramics substrates, shown in Figure 9, demonstrates their difference at high concentrations of the magnetic phase.

Comparison of the spectral dependences of the TKE for nanocomposites on different substrates also shows their significant differences (Figure 10).

At the concentration of *x*~21at.% in the nanocomposites on the glass-ceramics substrates, TKE values in the energy range E > 1.5 eV are approximately by 1.5–2 times less than in nanocomposites on the glass substrates, and the maxima are shifted to the low energy region. This may be connected with the different size of metal clusters due to different diffusion rates of the atoms on amorphous glass and more rough polycrystalline glass-ceramics substrates.

Position of the maximum values of MO effects and the trends of their change with an increase in alloy concentration greater than *x_fm_* are also different. For NC on the glass substrates, a wide maximum in the range of 1.5–2 eV (*x* = 20 at.%) gradually shifts to the region of 1.25 eV (at *x* = 47 at.%) with an increase of *x*. For NC on the glass-ceramics substrates for *x* = 21 at.% a maximum in the region of ~1.5 eV is observed, which gradually shifts to 2.0 eV (at *x* = 43 at.%) and at *x*= 47 at.% we can once again observe a maximum in the region of ~1.65 eV.

Figure 11 shows the concentration dependences of TKE for nanocomposites on glass-ceramics substrate at the different energy values. One maximum is observed in the region of *x* = 31 at.% for the energies of 1.97 and 3.17 eV, associated with the formation of hexagonal nanocrystals of the CoFeZr alloy. For the energy of 1.14 eV, a maximum is observed at the same concentration of *x* = 31 at.% and the beginning of the effect enhancement associated with the transition to a cubic BCC structure.

Comparison of the concentration dependences of TKE for nanocomposites on different substrates, made of glass and glass-ceramics, is shown in Figure 12. It is seen that the greatest divergence in the behavior of the concentration dependences is observed in the region of concentrations associated with the phase transition of CoFeZr nanocrystals from hexagonal to body-centered cubic structure.

The difference in the spectral dependences of TKE for nanocomposites (CoFeZr)_x_ (MgF_2_)_100−x_ on the glass and glass-ceramics substrates indicates that self-organization processes in nanocomposites on different substrates are different. The material and structure of the substrate (glass-amorphous, glass-ceramics–polycrystalline and more rough) affect the formation of CoFeZr nanocrystals with an increase in the concentration of the metal component *x* in the composites. The absence of the formation of a large negative maximum in the TKE spectra within the energy range of ~1.25 eV at the increase of *x* up to 46.4 at.% indicates that the transformation of CoFeZr nanocrystals from a hexagonal structure to BCC structure proceeds slower with *x* growth than in the nanocomposites on glass substrates.

The obtained results show that the processes of formation of CoFeZr alloy nanocrystals in the film composites of (CoFeZr)_x_ (MgF_2_)_100−x_, depending on the content of the metal component in the oxygen-free dielectric matrix, are associated with the substrate material and they affect magneto-optical properties of these nanocomposites.

## 5. Conclusions

Thus, for the first time, new complex investigations of the structural-phase transformations and magneto-optical properties of nanocomposites (CoFeZr)_x_ (MgF_2_)_100−x_ with a polyelement metal component in an oxygen-free dielectric matrix of magnesium fluoride on glass and glass-ceramics substrates have established the influence of the composition, substructure and phase transformations of nanocomposites on their magneto-optical properties.

MO spectroscopy revealed the features corresponding to phase structural transitions in magnetic granules during the transition from an amorphous CoFeZr alloy to a nanocrystalline CoFeZr alloy based on a hexagonal α-Co lattice and during the rearrangement of CoFeZr nanocrystals from a hexagonal to a cubic body-centered structure.

It was found that magnetic percolation threshold in nanocomposites (CoFeZr)_x_(MgF_2_)_100−x_ corresponds to the nominal value of *x**_fm_***~30 at.%, corresponding to the onset the formation of ferromagnetic nanocrystals of hexagonal syngony.

A comparison of the spectral and concentration dependences of TKE for NC on glass and glass-ceramics substrates showed that self-organization processes in nanocomposites on different substrates are different and the strongest differences occur in the region of the phase structural transition of CoFeZr nanocrystals from a hexagonal to a body-centered cubic structure at *x* = 38 at.% on the glass substrates and *x* = 46 at.% on glass-ceramics substrates.

## Figures and Tables

**Figure 1 nanomaterials-11-01666-f001:**
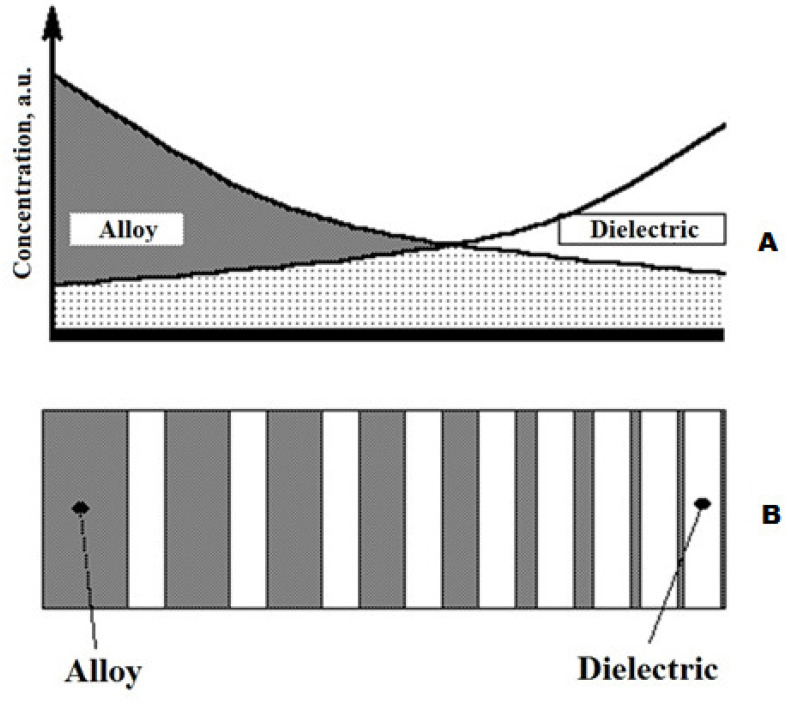
(**A**) A scheme for the distribution of the concentration of the composites targets components Co_45_Fe_45_Zr_10_ and MgF_2_ on the surface of the substrate. (**B**) A type of alloy target Co_45_Fe_45_Zr_10_ with dielectric inserts MgF_2_.

**Figure 2 nanomaterials-11-01666-f002:**
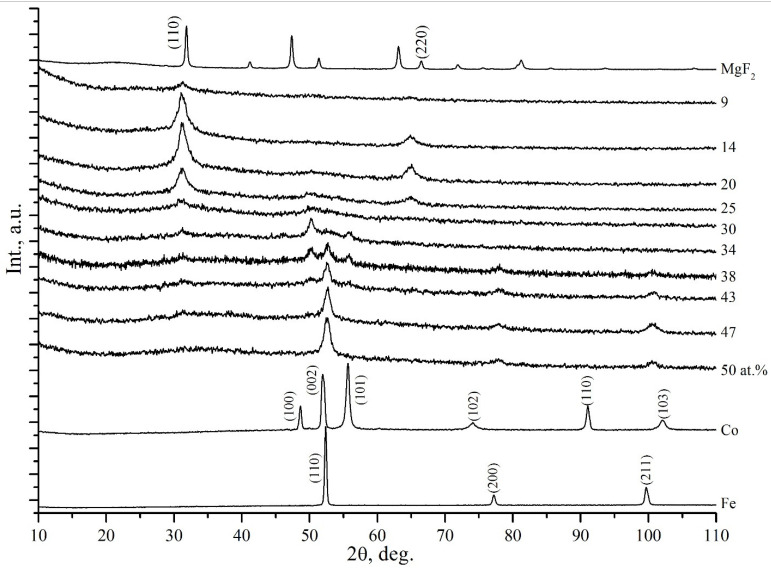
Diffractograms of nanocomposites with different compositions (CoFeZr)_x_ (MgF_2_)_100−x_ on the glass substrates.

**Figure 3 nanomaterials-11-01666-f003:**
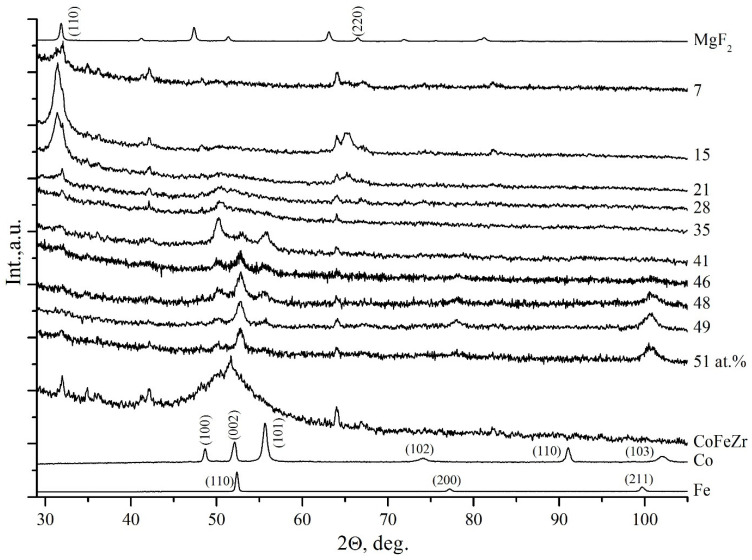
Diffraction patterns of nanocomposites of different composition (CoFeZr)_x_ (MgF_2_)_100−x_ on a glass-ceramics substrate.

**Figure 4 nanomaterials-11-01666-f004:**
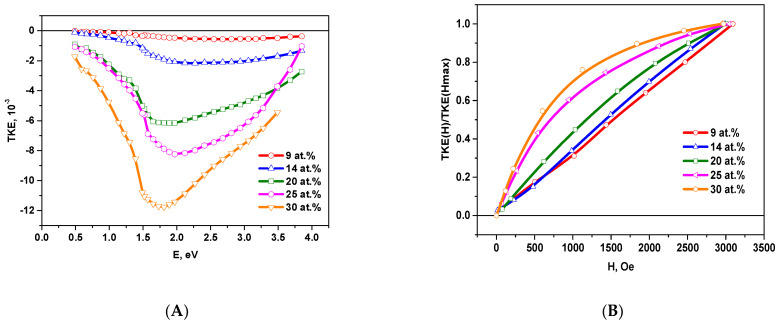
Spectral TKE(E) (**A**) and normalized field TKE(H)/TKE(Hmax) (**B**) dependences for composites on the glass substrates with metal phase concentrations *x* ≤ 30 at.%.

**Figure 5 nanomaterials-11-01666-f005:**
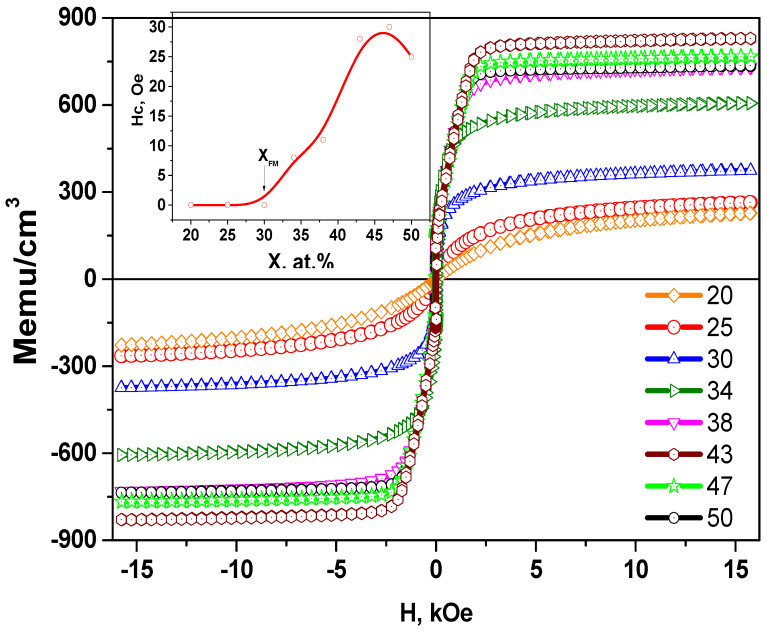
Hysteresis loops for nanocomposites (CoFeZr)_x_ (MgF_2_)_100−x_ on the glass substrates at different concentrations x, in the inset the dependences of the coercive force Hc on the metal concentration x in nanocomposites is presented.

**Figure 6 nanomaterials-11-01666-f006:**
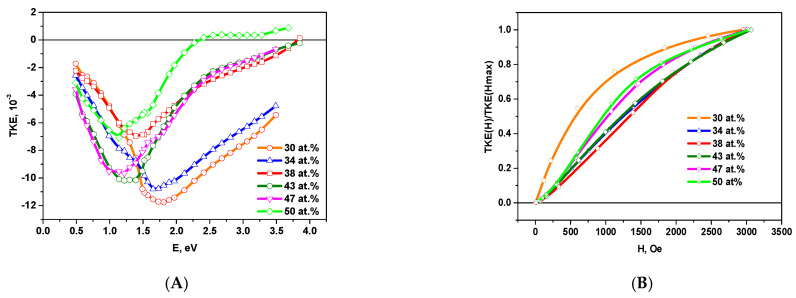
Spectral TKE(E) (**A**) and normalized field TKE(H)/TKE(Hmax) (**B**) dependences for composites on the glass substrates with metal concentrations *x* ≥ 30 at.%.

**Figure 7 nanomaterials-11-01666-f007:**
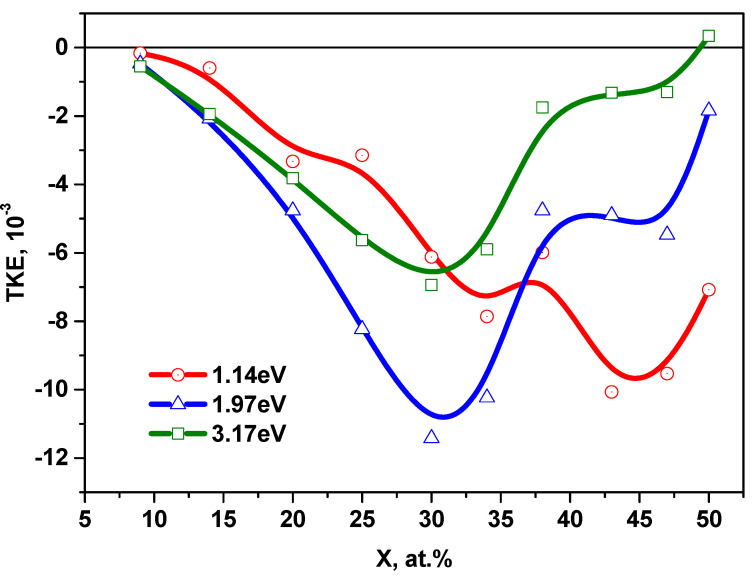
Concentration dependences of TKE of nanocomposites on the glass substrates at the different energy values.

**Figure 8 nanomaterials-11-01666-f008:**
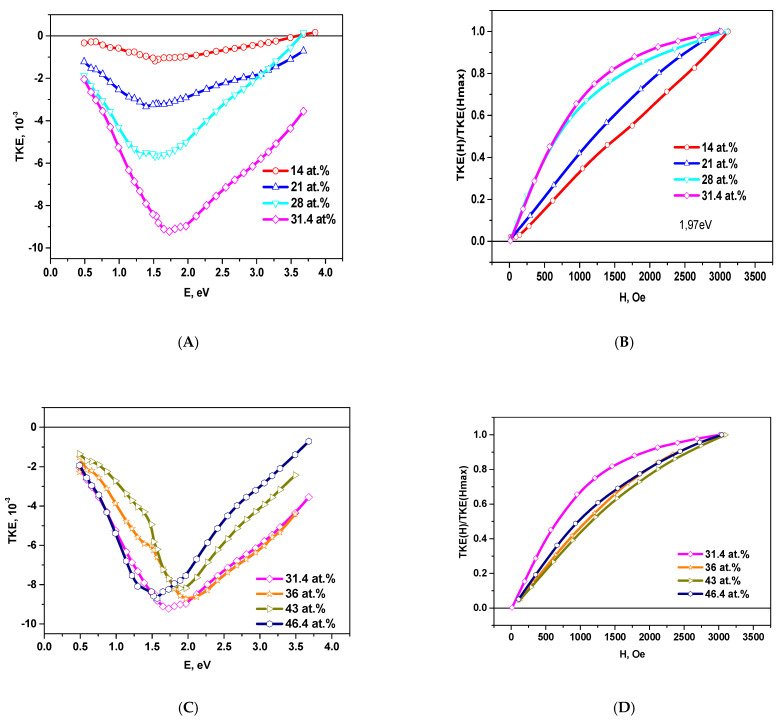
Spectral TKE(E) (**A**,**C**) and normalized field TKE(H)/TKE(Hmax) (**B**,**D**) dependences for nanocomposites on the glass-ceramics substrates with metal concentration less and greater *x* = 31.4 at.%.

**Figure 9 nanomaterials-11-01666-f009:**
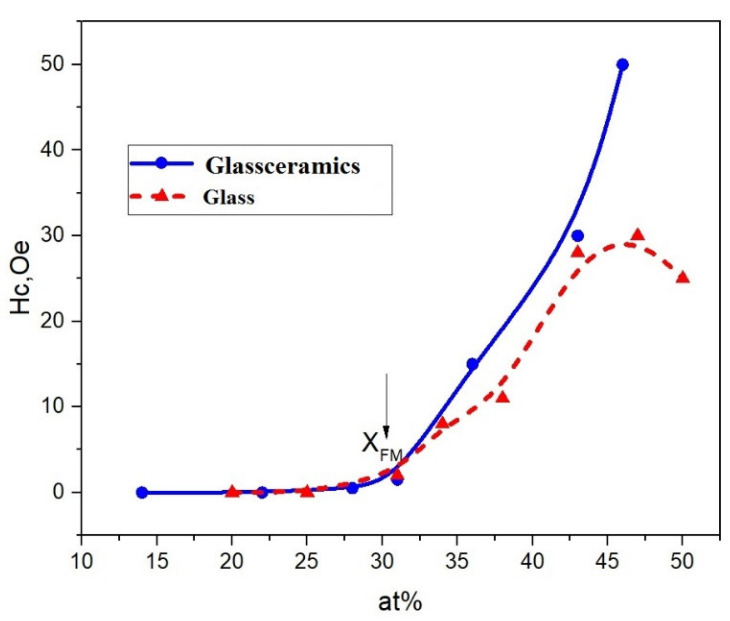
Dependences of the coercive force H_c_ on the concentration of metal alloys in nanocomposites (CoFeZr)_x_ (MgF_2_)_100−x_ on glass-ceramics and glass substrates.

**Figure 10 nanomaterials-11-01666-f010:**
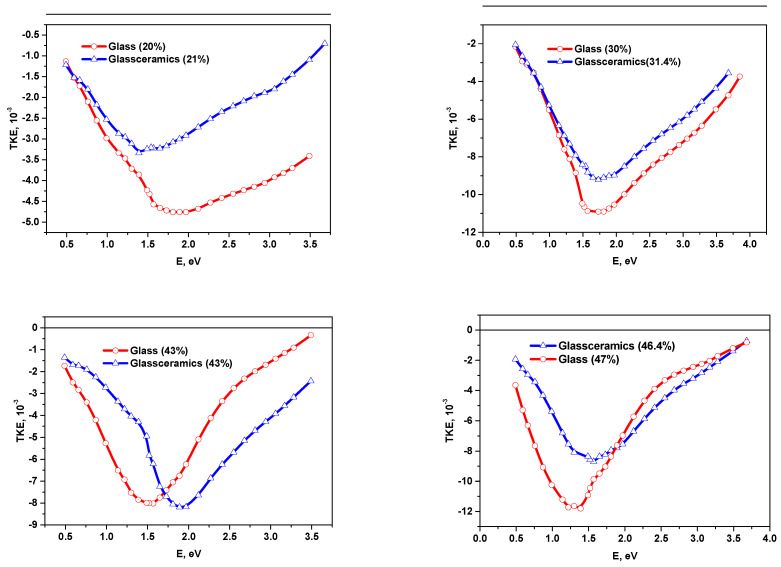
Comparison of the spectral dependences of TKE for nanocomposites (CoFeZr)_x_ (MgF_2_)_100−x_ on glass and glass-ceramics substrates.

**Figure 11 nanomaterials-11-01666-f011:**
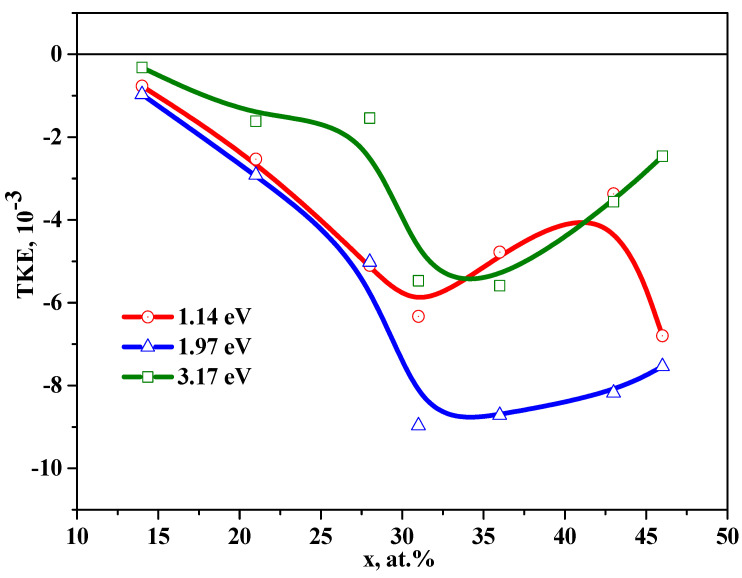
Concentration dependences of TKE for nanocomposites (CoFeZr)_x_ (MgF_2_)_100−x_ on the glass-ceramics substrates for different energy values.

**Figure 12 nanomaterials-11-01666-f012:**
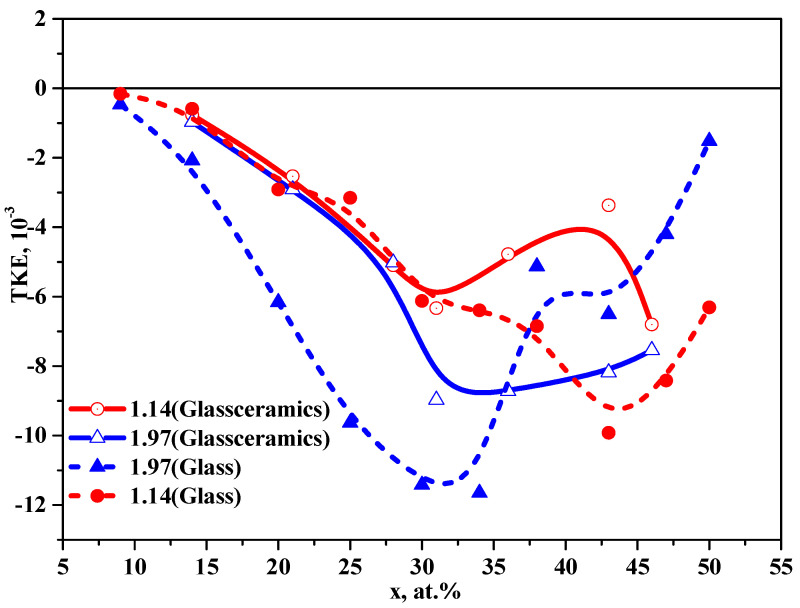
Comparison of the concentration dependences of TKE for nanocomposites (CoFeZr)_x_ (MgF_2_)_100−x_ on the glass and glass-ceramics substrates.

## Data Availability

Data available in a publicly accessible repository.

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
