# Peer review of "Effect of Phase Transformations of a Metal Component on the Magneto-Optical Properties of Thin-Films Nanocomposites (CoFeZr)x (MgF2)100−x"

_nanomaterials, 2021, doi:10.3390/nano11071666_

Round 1
Reviewer 1 Report
The manuscript does not follow the Guidelines. Also, the quality and organization of the text and figures is weak. More care should be taken to present the results and facilitate understanding of the work. The conducted research could be convincingly (or not) when presented in an organized/ systematic way.
Please sharpen the description of the novelty factor in the "in this work" section of the introduction. What exactly was done in this study for the first time? The significance of this study should be more emphasize in the introduction.
The material synthesis section is written too general as a kind of report, and not suitable for a scientific paper. It should be rewritten in a concise and concert form including deep discussion. The experimental details of the tests are not sufficient. Details must be given. Also, the information about the supplier and purity of the starting materials is highly desirable.
Results and discussion. To increase the scientific value of the manuscript, specific comparisons should be made to previously published materials that have a similar purpose. Also, please present a strong case for how this work is a major advance. This needs to be done in the manuscript itself, not just in the response to review comments.
The conclusions section needs to be shorten/ revised in order to highlight the main findings. In this form, this chapter there is no summary of all significant obtained results. Also, please explain the specific ways in which this work fundamentally advances the field relative to prior literature.
Please check the manuscript and refine the language carefully.
Few references and a tendency of self-citation is remarked. This should be reconsidered.
Finally, I consider that the work is not fit for publication in this form and needs large revision. If the paper will not be improved/ revised accordingly, I will not recommend the publication.
Author Response
Comments from reviewer 1
- Please sharpen the description of the novelty factor in the "in this work" section of the introduction. What exactly was done in this study for the first time? The significance of this study should be more emphasize in the introduction.
Response:
We have inserted in the section 1. Introduction:
Thus , for the first time, new data will be presented on the magneto-optical properties and the effect of phase transitions on them when changing the composition of a composite obtained of a three-element alloy CoFeZr and an oxygen-free matrix MgF2 and the MO percolation threshold will be determined.
2.The material synthesis section is written too general as a kind of report, and not suitable for a scientific paper. It should be rewritten in a concise and concert form including deep discussion. The experimental details of the tests are not sufficient. Details must be given. Also, the information about the supplier and purity of the starting materials is highly desirable.
Response:
We have inserted new text and new Fig.1 in the section 2. Objects and methods of the study:
An alloy target of the composition Co45Fe45Zr10 was used to precipitate the amorphous ferromagnetic metal component of the composites. It was made by vacuum melting from metals of the appropriate composition using an induction furnace. Preparation of attachments for the alloy was carried out from carbonyl high-purity iron (99.9 %), high-purity cobalt (99.98 %) and zirconium (99.8 %) with a weight content of components in accordance with the composition of the alloy Co45Fe45Zr10. The melt of the corresponding composition was poured into a specially prepared ceramic mold in a vacuum. The target had sizes of 270×70×14 mm. The target were ground on both sides, soldered to a water-cooled base, and placed in the spray position in the ion-beam sputtering facility. MgF2 insets with a thickness of ~ 2 mm and a width of ~ 9 mm were fixed on the surface of the alloy target perpendicular to its longitudinal axis, not equidistant, in order to obtain a continuous set of concentrations of the components of the composite target along its length in one technological deposition cycle (Fig. 1).
For the production of dielectric inserts, magnesium fluoride MgF2 for optical ceramics, doped with calcium, pure (GOST 6-09-01-731-91) with a molecular weight of 62.30 atomic mass units was used. The mass fraction of calcium (Ca) was about 0.1 - 0.3%.
Figure 1. (a) - A scheme for the distribution of the concentration of the composites targets
components Co45Fe45Zr10 and MgF2 on the surface of the substrate.
(b) - A type of alloy target Co45Fe45Zr10 with dielectric inserts MgF2.
Deposition was performed on glassceramics and glass substrates with a size of 60x48 mm2. The use of non-uniform arrangement of dielectric inserts allowed us to obtain a continuous spectrum of changes in the compositions of alloy and dielectric components of the composite from 20 to 70 at % of the metal phase in one technological cycle of deposition. The approximate ratio of the volume fraction of an alloy to the volume fraction of a dielectric with a uniform distribution of attachments can be estimated by the formula:
Vf/Vd = (Rf/Rd) * (Sf/Sd) = (Rf/Rd) * (S – ns) / ns = (Rf/Rd) * (L – nb) / nb
where: RF and RD - coefficients spraying metal alloy and dielectric, respectively; SF and SD sputtering target area occupied by the alloy and dielectric, respectively; S is the area of the entire target (270 x 70 mm2); n - the number of dielectric inserts in composite target; s - area of a single insert (9 x 70 mm2); L - length of the target compound (270 mm); b - width of the dielectric insert.
- Results and discussion. To increase the scientific value of the manuscript, specific comparisons should be made to previously published materials that have a similar purpose. Also, please present a strong case for how this work is a major advance. This needs to be done in the manuscript itself, not just in the response to review comments.
Response:
All the results presented in sections 3 and 4 Results and discussion are new. However, we have added the Figure 5 and the corresponding text to it.
Figure 5. Hysteresis loops for nanocomposites(CoFeZr)x(MgF2)100-x on the glass substrates at different concentrations х, in the inset the dependencs of the coercive force Hc on the metal concentration x in nanocomposites is presented.
Text: Analysis of the hysteresis loops showed that the value of the coercive force Hc in all of the composites at x≤30 at.% is equal zero. Therefore the main contribution to the coercive force is made by single superparamagnetic particles, and after that, a nonlinear increase in the coercive force is observed, which is explained by the formation of FM clusters in the sample
- The conclusions section needs to be shorten/ revised in order to highlight the main findings. In this form, this chapter there is no summary of all significant obtained results. Also, please explain the specific ways in which this work fundamentally advances the field relative to prior literature.
Response:
The new text of 5. Conclutions:
- Thus , for the first time, new complex investigations of the structural-phase transformations and magneto-optical properties of nanocomposites (CoFeZr)x(MgF2)100-x with a polyelement metal component in an oxygen-free dielectric matrix of magnesium fluoride on glass and glassceramics substrates have established the influence of the composition, substructure and phase transformations of nanocomposites on their magneto-optical properties.
- MO spectroscopy revealed the features corresponding to phase structural transitions in magnetic granules during the transition from an amorphous CoFeZr alloy to a nanocrystalline CoFeZr alloy based on a hexagonal α-Co lattice and during the rearrangement of CoFeZr nanocrystals from a hexagonal to a cubic body-centered structure.
- It was found that magnetic percolation threshold in nanocomposites (CoFeZr)x(MgF2)100-x corresponds to the nominal value of xfm~ 30 at%, corresponding to the onset the formation of ferromagnetic nanocrystals of hexagonal syngony.
- A comparison of the spectral and concentration dependences of TKE for NC on glass and glassceramics substrates showed that self-organization processes in nanocomposites on different substrates are different and the strongest differences occur in the region of the phase structural transition of CoFeZr nanocrystals from a hexagonal to a body-centered cubic structure at x=38 at% on the glass substrates and x=46 at% on glassceramics substrates.
5.Please check the manuscript and refine the language carefully.
Response:
We checked the manuscript.
- Few references and a tendency of self-citation is remarked. This should be reconsidered.
Response:
We added 4 new papers in References
[1] Sankar S., Berkowitz A.E.,Smith D. J., Spin-dependent transport of Co-SiO2 granual films approaching percolation, Phys.Rev. B,62,No 21(2000) 14237-14278.
[2] Mitani S.,Fudjimori H.,Ohnuma S., Spin-dependent tunneling phenomena in insulating granular systems, JMMM,165 (1997) 141-148.
[3] Dongsheng Yao, Shihui Ge, Bangmin Zhang, Huaping Zuo, Xueyun Zhou,
Fabrication and magnetism of granular films for high frequency application,J. Appl. Phys. 103 (2008), 113901-113905; doi: 10.1063/1.2932076.
[4] Dongsheng Yao, Shihui Ge, Bangmin Zhang, Huaping Zuo, Xueyun Zhou, Fabrication and magnetism of granular films for high frequency application, J. Appl. Phys. 103 (2008) 113901 (2008; doi: 10.1063/1.2932076

Reviewer 2 Report
This paper presents the synthesis method and various characterization methods for (CoFeZr)x (MgF2)1-x nanocomposites using different physics effects, which worth publication.
However, English and physics incorrectnesses (to mention few) should be improved in the following lines:
l. 59: "It can be states..."(typo) -> It can be stated
l. 63: inherent in -> inherent to (would be better)
ls. 71-72: "The electrical conductivity in this composite is determined by the hopping conductivity of the electrons that tunnel from one metal particle to another through the dielectric regions." Please revise; as far as I know, the hopping conductivity and tunneling conductivity are not the same.
l. 74-75: "Unusual transport phenomena, such as the tunnel anomalous Hall effect and the logarithmic temperature dependence of the conductivity, were found in this concentration region..." Please revise; dependence is not a phenomenon
Author Response
Comments from reviewer 2
1.However, English and physics incorrectnesses (to mention few) should be improved in the following lines:
- 59: "It can be states..."(typo) -> It can be stated
- 63: inherent in -> inherent to (would be better)
Response: We have corrected the incorrectness in English as much as possible.
2 ls. 71-72: "The electrical conductivity in this composite is determined by the hopping conductivity of the electrons that tunnel from one metal particle to another through the dielectric regions." Please revise; as far as I know, the hopping conductivity and tunneling conductivity are not the same
Response We have fixed an incorrect phrase about tunneling conduction on:
With decreasing metal concentration x, from x = 100 at.% down to the percolation threshold Xper, the resistivity of nanocomposites increases, as physical contact between most of the granules continuously disappears. There is still the tunneling conductivity for concentrations down to Xc (Xper >X>Xc), where the metal–insulator transition occurs. In this concentration range, novel transport properties, such as tunnel anomalous Hall effect and logarithmic temperature dependence of conductivity, have been discovered [13]. Hopping conductivity of different types as well as cotunneling was observed in dielectric regimes below Xc [12,14,15]
3 l. 74-75: "Unusual transport phenomena, such as the tunnel anomalous Hall effect and the logarithmic temperature dependence of the conductivity, were found in this concentration region..." Please revise; dependence is not a phenomenon
Response We replaced «Unusual transport phenomena» - « Unusual transport properties»

Reviewer 3 Report
This is well written, well organized manuscript presenting mainly magneto-optical properties of metallic nanocomposites with respect to the concentration of metal component. The data are present well and adequately discussed.
The only concern of mine is that the authors didn’t present magnetic hysteresis loops of the measured samples, which can be a benefit for the publication, since the amplitude of the measured Kerr effect is proportional to the magnetization. If the applied magnetic field is not high enough to magnetically saturate the sample, one should be careful to comment the amplitude dependence of the Kerr effect. Showing magnetic hysteresis loops, and therefore describing magnetic properties of the samples, could clarify this.
Author Response
Comments from reviewer 3
The only concern of mine is that the authors didn’t present magnetic hysteresis loops of the measured samples, which can be a benefit for the publication, since the amplitude of the measured Kerr effect is proportional to the magnetization. If the applied magnetic field is not high enough to magnetically saturate the sample, one should be careful to comment the amplitude dependence of the Kerr effect. Showing magnetic hysteresis loops, and therefore describing magnetic properties of the samples, could clarify this.
Response
We have added hysteresis loops data in Figure 5.

Round 2
Reviewer 1 Report
Authors addressed referee's comments and improved their manuscript that is publishable in this form.
Author Response
We have improved the English Language and corrected the numbers of some references to literature.
